# Effects of a dietary intervention on cardiometabolic risk and food consumption in a workplace

**Archana Shrestha**[1,2]*, **Dipesh Tamrakar**[3], **Bhawana Ghinanju**[1], **Deepa Shrestha**[1], **Parashar Khadka**[1], **Bikram Adhikari**[1], **Jayana Shrestha**[4], **Suruchi Waiwa**[1], **Prajjwal Pyakurel**[5], **Niroj Bhandari**[6], **Biraj Man Karmacharya**[1], **Akina Shrestha**[1], **Rajeev Shrestha**[7], **Rajendra Dev Bhatta**[8], **Vasanti Malik**[9,10], **Josiemer Mattei**[9], **Donna Spiegelman**[11]

1 Department of Public Health, Kathmandu University School of Medical Sciences, Dhulikhel, Bagmati, Nepal, 2 Department of Chronic Disease Epidemiology, Yale School of Public Health, New Haven, Connecticut, United States of America, 3 Department of Community Medicine, Kathmandu University School of Medical Sciences, Dhulikhel, Bagmati, Nepal, 4 Department of Physiotherapy, Kathmandu University School of Medical Sciences, Dhulikhel, Bagmati, Nepal, 5 Department of Community Medicine, BP Koirala Institute of Health Sciences, Dharan, Koshi, Nepal, 6 Department of Medicine, Kathmandu University School of Medical Sciences, Dhulikhel, Bagmati, Nepal, 7 Department of Pharmacology, Kathmandu University School of Medical Sciences, Dhulikhel, Bagmati, Nepal, 8 Department of Biochemistry, Kathmandu University School of Medical Sciences, Dhulikhel, Bagmati, Nepal, 9 Department of Nutrition, Harvard TH Chan School of Public Health, Boston, Massachusetts, United States of America, 10 Department of Nutritional Sciences, Faculty of Medicine, University of Toronto, Toronto, Ontario, Canada, 11 Department of Biostatistics and Center of Methods for Implementation and Prevention Sciences, Yale School of Public Health, New Haven, Connecticut, United States of America

* archana@kusms.edu.np

## Abstract

### Background

Worksite-based health programs have shown positive impacts on employee health and have led to significant improvements in cardiovascular risk factor profiles. We aimed to determine the effect of cafeteria intervention on cardio-metabolic risk factors diet in a worksite setting (Dhulikhel Hospital) in Nepal.

### Methods

In this one-arm pre-post intervention study, we recruited 277 non-pregnant hospital employees aged 18–60 with prediabetes or pre-hypertension. The study was registered in clinicaltrials.gov (NCT03447340; 2018/02/27). All four cafeterias in the hospital premises received cafeteria intervention encouraging healthy foods and discouraging unhealthy foods for six months. We measured blood pressure, fasting glucose level, glycated hemoglobin, cholesterol in the laboratory, and diet intake (in servings per week) using 24-hour recall before and six months after the intervention. The before and after measures were compared using paired-t tests.

**Data Availability Statement:** All relevant data are within the manuscript and its Supporting information files.

**Funding:** The study was funded by National Institute of Health (NIH) Director's Pioneer Award (Award #DP1ES025459). The funders had no role in study design, data collection and analysis, decision to publish, or preparation of the manuscript.

**Competing interests:** The authors have declared that no competing interests exists.

## Results

After six months of cafeteria intervention, the median consumption of whole grains, mono/polyunsaturated fat, fruits, vegetable and nuts servings per week increased by 2.24 (p<0.001), 2.88(p<0.001), 0.84(p<0.001) 2.25(p<0.001) and nuts 0.55 (p<0.001) servings per week respectively. The median consumption of refined grains decreased by 5.07 servings per week (p<0.001). Mean systolic and diastolic blood pressure decreased by 2 mmHg (SE = 0.6; p = 0.003) and 0.1 mmHg (SE = 0.6; p = 0.008), respectively. The low-density lipoprotein (LDL) was significantly reduced by 6 mg/dL (SE = 1.4; p<0.001).

## Conclusion

Overall, we found a decrease in consumption of refined grains and an increase in consumption of whole grains, unsaturated fats, fruits, and nuts observed a modest reduction in blood pressure and LDL cholesterol following a 6-month cafeteria-based worksite intervention incorporating access to healthy foods.

## Introduction

Cardiovascular diseases (CVD) are a significant cause of disability and premature death, accounting for one-third of total deaths globally each year [1, 2]. Approximately 80% of these deaths occur in low and middle-income countries (LMIC) [3, 4]. Global costs attributable to CVD were estimated at $863 billion in 2010, with 55% due to direct healthcare costs and 45% due to loss of productivity from disability and premature death or time lost from work due to illness. These costs are projected to increase to $1,044 billion by 2030 [5, 6]. In LMIC, it was estimated that nearly 2% of the Gross Domestic Product (GDP) was lost due to CVD between 2011 and 2015 [7].

A sedentary lifestyle, poor diet, and excessive body weight are well-established major risk factors for CVD [8, 9]. The evidence for the ability of lifestyle interventions to improve diet and increase exercise to reduce cardiovascular risk is substantial [10, 11]. Despite the evidence supporting the use of lifestyle interventions to prevent CVD, their translation into real-world settings has been challenging. Both individual and group-level interventions have effectively promoted behavior change; however, attrition rates in these programs are high, and behavior change is typically not sustained [12, 13]. Social ecological approaches that address social and environmental factors to prevent cardiometabolic risk reach broadly to target large segments of the population rather than focusing on a small number of selected individuals [14, 15]. Because many people spend most of their time at work, worksites provide unique opportunities for health promotion and disease prevention programs, (24) and worksites have a number of characteristics that could support multi-component, multi-level interventions. For example, worksites typically feature existing social networks, formal communication systems, readily available eating environments, and access to employer support programs.

Worksite-based health programs have positively impacted employee health [16, 17] and significantly improved cardiovascular risk factor profiles [18]. Worksite interventions encompassing environmental changes (i.e., low-cost healthy food options or spaces for physical activity, such as gyms, have been highlighted as components of successful worksite interventions [18, 19]. Previous studies that evaluated environmental level changes in worksite settings have shown effects for preventing obesity [20, 21], and lowering lipids [20–22], blood glucose

levels [20–22], and blood pressure [14, 23, 24]. However, few studies have evaluated an independent effect of environmental intervention, while others examined the combined effect of environmental and education interventions [8, 10, 15]. Moreover, much of the evidence comes from high-income countries, and there is limited data from low-income settings. The Nepal Pioneer Worksite Intervention Study [25] aims to assess the effectiveness of a workplace environmental, dietary modification intervention and a nutrition education intervention both alone and in combination. In this paper, we report the changes in cardiovascular risk factors and dietary consumption (servings per week) among high-risk employees in a worksite setting after six months of an environmental dietary modification.

## Methods

### Study design

We used a single-arm pre-post study design to assess how cafeteria intervention providing a healthier diet affects diet consumption and cardiometabolic risk. The institutional review committee at Harvard T.H. Chan School of Public Health, Nepal Health Research Council, and Kathmandu University School of Medical Sciences has approved the study protocol. The study was registered in clinicaltrials.gov (Identification Member: NCT03447340) on February 27, 2018).

### Study setting

The study was conducted at Dhulikhel Hospital—Kathmandu University Hospital (DH-KUH) in central Nepal. Dhulikhel Hospital is an independently owned, not-for-profit institution conceived and supported by the Dhulikhel community. The hospital has approximately 1040 employees and four cafeterias operating 16 hours daily. There are no gyms or other fitness facilities to promote physical activity onsite. The hospital does not currently offer any wellness programs to prevent cardiometabolic diseases.

### Participant recruitment

We conducted a screening of participants to identify eligible participants. The inclusion criteria were: (a) adults 18 years or older; (b) full-time employees of DH-KUH; with (c) untreated pre-diabetes and/or pre-hypertension or hypertension, defined as systolic blood pressure of 120 mmHg or diastolic pressure 80 mmHg; or glycated hemoglobin (HbA1c) of 5.7% to 6.4%, or fasting blood sugar of 100 mg/dL. The participants were recruited from May 2018 to July 2020. The exclusion criteria were: (a) pregnant women, since dietary habits may change during pregnancy, (b) taking diabetes medication, or (c) taking hypertension medication. In the first phase of screening, the research assistants measured blood pressure and administered the Indian Diabetes Risk Score (IDRS) [26, 27]. The IDRS considers age, abdominal obesity, self-reported physical activity, and family history of diabetes to calculate a score from 0 to 100, with a higher number indicating higher risk. The IDRS has been considered a reliable instrument to screen the risk of diabetes in Asian-Indian populations [26–29]. In the second screening, the participants scoring 30 or more on IDRS were asked to provide a blood sample to measure HbA1c and fasting blood glucose. Individuals with HbA1c between 5.7% and 6.4% or fasting blood glucose of 100 mg/dL or more were invited to participate. After six months of screening, we invited all employees for screening and recruited 277 who met the inclusion criteria after providing written consent. No compensation or reimbursement was supplied to the participants.

## Intervention

We took baseline measurements before starting the cafeteria intervention. Then after, the participants received the cafeteria intervention for six months. The details of the intervention are published elsewhere [25]. Briefly, the four cafeterias in the hospital improved the quality of their meals by a) increasing the availability of fresh whole fruit (not fruit juice) and vegetable options, b) avoiding sales of sugar-sweetened beverages and foods such as pastries, cookies, and cream donuts, c) replacing refined grains with whole grains in cooking; d) continue using healthy vegetable oils such as soy and sunflower; e) minimizing the sale of fried foods; f) trimming animal fats from meats before cooking; g) using healthier protein sources, such as chicken, beans, and nuts instead of red meat(8) making potable water free of cost; and h) reducing salt in cooking. These guidelines were based on the recommendations for a healthy diet to improve cardiovascular health [30]. To facilitate these changes, a cafeteria operation team was formed and trained on procedures to implement, supervise, and monitor the healthy changes. In addition, we trained the cafeteria staff on healthy eating and recipe modifications to incorporate healthy options.

## Outcomes

The outcomes were

(a) Cardiometabolic risks: absolute changes in HbA1c, systolic blood pressure, diastolic blood pressure, Body Mass Index (BMI), High-Density Lipoprotein (HDL), Low-Density Lipoprotein (LDL), triglycerides, and total cholesterol at two endpoints–before and after cafeteria intervention.

(b) Diet consumption: absolute change in food intake in servings per week (whole grains, refined grains, potato, vegetables, fruits, Fats Mono/Poly, saturated fats, lentils, nuts, red meat, white meat, fish, soda drinks and other beverages (Tea/ Coffee)) before and after the cafeteria intervention.

## Data collection

**Blood pressure.** Trained research assistants (RA) measured blood pressure in the right arm of seated participants after a five-minute rest period. Three systolic and diastolic blood pressure measurements were taken using a Microlife automatic blood pressure measuring device. The mean of three blood pressure measurements was employed for the analysis.

**Anthropometry.** Body weight was measured with minimum clothing and without shoes using an Omron Model HBF-400 scale and recorded to the nearest 0.1 pounds. The weighing scales were calibrated to zero every day. Participants' heights were measured without shoes while standing against a wall. Height was measured using tape and recorded to the nearest 0.1cm. BMI was calculated as the weight (kilograms) divided by the square of the height (meters).

**Laboratory measurements.** Blood samples were analyzed for HbA1c, fasting glucose, LDL cholesterol, HDL cholesterol, triglycerides, and total cholesterol. All the laboratory assays were carried out in the biochemistry laboratory of DH-KUH. Blood samples were collected using evacuated blood collection tubes. Participants were asked to fast overnight (8–14 hours). HbA1c was measured using the Boronate affinity chromatography [31, 32]; fasting blood glucose using Hexokinase method [33]; LDL and HDL using the elimination method [34]; triglyceride using GPO-PAP [35]; and total cholesterol using CHOD-PAP [34, 35]. For each type of assay, the laboratory had quality control (QC) materials (using commercially available assayed

and unassayed control material) from BIORAD Laboratories, USA. Each QC was run at least in duplicate. The laboratory routinely performed the External Quality Assurance Scheme from unknown assayed samples from the Department of Clinical Biochemistry CMC, Vellore, India, for 23 routine parameters, 5 immunological parameters, and HbA1c.

**Food consumption.** To measure dietary intake, we conducted two interviewer-administered 24-hour dietary recalls within a week. Each 24-hour dietary recall takes approximately 25 min to complete. First, the previous day's activities were documented to refresh the participant's memory. Then, a trained research assistant asked the participant to recall everything s/he consumed from the first to the last. The time and place of each meal were noted, followed by detailed information on each food, including specific brands, ingredients, and/or recipes. Participants were asked to report their food portions using colorful examples of sizes or household measures such as spoons, bowls, etc. If a participant reported using their recipe, then complete information on each ingredient would be inquired about. The dietary intake was categorized into food groups (whole grains, refined grains, potato, vegetable, fruits, fats (mono/polyunsaturated), fat (saturated), nuts, red meat, white meat, lentil, and fish) (supplementary file) and food groups servings per week were estimated.

**Socio-demographic and lifestyle assessment.** Trained RAs interviewed the participants using a standardized electronic questionnaire using Open Data Kit software [36]. The questionnaire assessed socioeconomic characteristics, including age (in years), sex (male/female), ethnicity (brahmin-chhetri/ newar/ tamang/ others), religion (hindu/non-hindu), education (number of formal years of education), and lifestyle factors including smoking (never/former/current), alcohol intake (drinks per day), and physical activity (MET minutes per week). We used the Global Physical Activity Questionnaire [37] to estimate metabolic equivalent of task (MET) minutes per week. A weekly MET equivalent of 600 would represent 30 minutes of brisk walking five times per week or 15 minutes of running five times per week.

## Data analysis

Sample characteristics were described for men and women using means and standard deviation (SD) for parametric numerical variables, median and interquartile range for non-parametric variables, and percentages for categorical variables. We applied a paired test to compare normally distributed variables like systolic blood pressure, diastolic blood pressure, BMI, HDL, HbA1c, and fasting blood glucose before and after intervention. We used paired Wilcoxon tests to compare non-normally distributed variables such as food groups (servings per week) before and after intervention. We compared the change in food group (servings per week), HbA1c (mg/dl), systolic blood pressure(mmHg), and lipids before and after cafeteria intervention and six months after using the paired t-test.

We utilized a linear mixed-effect model to determine and assess the difference in cardiometabolic risk factors, including HbA1c (%mg/dl), systolic and diastolic blood pressure(mmHg), and lipids (blood pressure (mg/dL) before and after the interventions, adjusting for age, gender, ethnicity, education, marital, and religion and participant unique identity as a random effect. We checked the assumptions of the random effects, which were found to follow a Gaussian distribution. We checked the Gauss-Markov assumption and performed a detailed residual analysis to ensure the reliability of the results. We used bootstrapping to estimate the confidence interval of the linear mixed-effect model parameters if the residuals were not normally distributed. We drew 1000 bootstrap samples from the original data set with replacement and computed the 95% confidence intervals using the "boot" package. We used R version 4.3.1 to perform pre-analytical processing and statistical analysis.

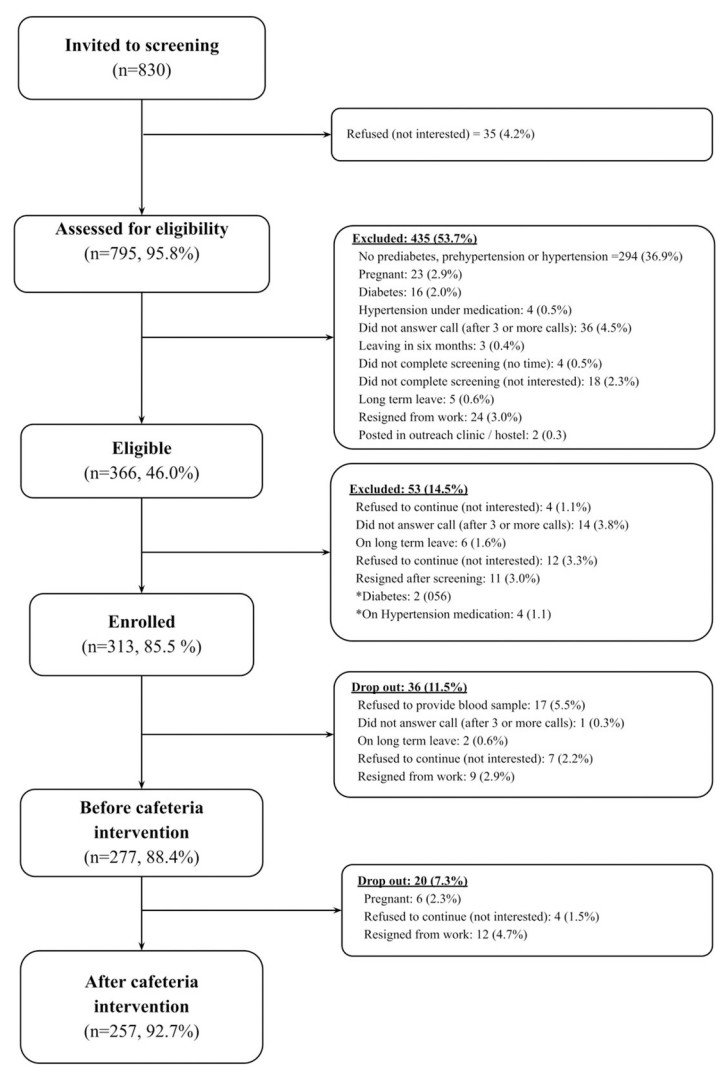

**Fig 1. Study flow diagram.**

## Results

### Study enrollment and follow-up

We invited 830 employees for screening, of which 35 (4%) refused. The remaining 795 employees attended the screening. From these, 435 (55%) were excluded (337 were ineligible, 58 were unwilling, and 34 were for other reasons (Fig 1). Of the remaining 360 employees, 313 (86%) participated in the study. Thirty-six (13%) dropped out (25 were unwilling, 11 resigned, or were on long-term leave) before the start of the cafeteria intervention. Hence, 277 participants received cafeteria intervention. After 6 months of cafeteria intervention, 20 (7%) dropped out (4 unwilling, 10 resigned, 6 pregnant). Fig 1 shows the details of study enrollment and follow-up. For the analysis of food groups, complete data was available for 245 participants.

### Characteristics of participants

The characteristics of the 277 participants (155 women and 122 men) at baseline are shown in Table 1. The mean age of the participants was 32.3 (SD = 7.9) years. Half of the participants

Table 1. Characteristics of the study participants at baseline (n = 277).

| Characteristics | Women | Men | Total |
|---|---|---|---|
| | (n = 155) | (n = 122) | (n = 277) |
| | n (%) | n (%) | n (%) |
| Age, years Mean (SD) | 31.0 (8.6) | 33.8 (6.8) | 32.3 (7.9) |
| Ethnicity | | | |
| Brahmin/Chhetri | 55 (35.5) | 42 (34.4) | 97 (35.0) |
| Newar | 80 (51.6) | 58 (47.5) | 138 (49.8) |
| Tamang | 15 (9.7) | 12 (9.8) | 27 (9.7) |
| Others | 5 (3.2) | 10 (8.2) | 15 (5.4) |
| Marital Status | | | |
| Married | 99 (63.9) | 96 (78.7) | 195 (70.4) |
| Not Married | 56 (36.1) | 26 (21.3) | 82 (29.6) |
| Religion | | | |
| Hindu | 137 (88.4) | 110 (90.2) | 247 (89.2) |
| Not Hindu | 18 (11.6) | 12 (9.8) | 30 (10.8) |
| Education | | | |
| Less than high school | 32 (20.6) | 48 (39.3) | 80 (28.9) |
| High School or more | 123 (79.4) | 74 (60.7) | 197 (71.1) |
| Years of education, Mean (SD) | 12.9 (3.92) | 12.5 (4.06) | 12.7 (3.98) |

(52%) were from the Newars, and 35% were Brahmin/Chhetri. The majority were Hindu (89%). More women had a high-school education compared to men.

The distribution of CVD risk factors in the study participants at baseline is shown in Table 2. More men were smokers, drank alcohol, had quite high alcohol intake on average, and had higher levels of physical activity compared to women. More men had high blood pressure. More women were classified as overweight and obese compared to men.

The proportion of food intake in servings per week before and after the cafeteria intervention is presented in Table 3. Six months post cafeteria intervention, the median consumption of whole grains, mono/polyunsaturated fat, fruits increased, and nuts increased, whereas the median consumption of refined grains decreased. There were no significant changes in the consumption of potatoes, vegetables, saturated fats, lentils, red meat, SSB, and white meat.

The CVD risk factors and biomarkers before and after cafeteria intervention are presented in Table 4.

The mean systolic blood pressure and diastolic blood pressure decreased significantly by 1.69 mm of Hg (95% CI: -2.81, -0.57; p = 0.003) and 1.08 mm of Hg (95% CI: -1.89, -0.33; p = 0.01) respectively after cafeteria intervention after adjusting for potential confounders such as age, gender, education, ethnicity, and religion. As shown in Table 4, the Mean LDL decreased significantly by 6.71 mg/dL (95% CI:-9.53, -3.89; p<0.001) after the cafeteria intervention after adjusting for confounders. The mean fasting blood sugar increased by 4.42 mg/dL (95% CI:3.11, 5.87; p<0.001) during the cafeteria intervention period in the linear mixed effect model adjusted for potential confounders such as age, gender, education, ethnicity, and religion. The mean HbA1c remained unchanged during cafeteria intervention. There were no significant changes in BMI, triglycerides, and total cholesterol.

## Discussion

In a worksite in central Nepal, six months of cafeteria intervention to facilitate the intake of healthy foods and discourage the increase of unhealthy food resulted in positive changes in

**Table 2. Cardiovascular risk factors of the study participants at baseline (n = 277).**

| Risk factors | Women (n = 155) | Men (n = 122) | Total (n = 277) |
|---|---|---|---|
| | n (%) | n (%) | n (%) |
| **Smoking** | | | |
| Never | 151 (97.4) | 66 (54.1) | 217 (78.3) |
| Former | 4 (2.6) | 15 (12.3) | 19 (6.9) |
| Current | 0 (0.0) | 41 (33.6) | 41 (14.8) |
| **Alcohol intake (drinks per day)** | | | |
| Never | 101 (65.2) | 33 (27.0) | 134 (48.4) |
| 1 or less drinks per day | 47 (30.3) | 40 (32.8) | 87 (31.4) |
| 1–2 drinks per day | 3 (1.9) | 7 (5.7) | 10 (3.6) |
| 3 or more drinks per day | 4 (2.6) | 41 (33.6) | 45 (16.2) |
| **Physical activity** | | | |
| Low (<600 METmin/ week) | 71 (45.8) | 38 (31.1) | 109 (39.4) |
| High (600 or more METmin/week) | 84 (54.2) | 84 (68.9) | 168 (60.6) |
| **Body Mass Index** | | | |
| Underweight (18.5 kg/m2 or less) | 76 (49.0) | 59 (48.4) | 135 (48.7) |
| Normal (19.5–24.9 kg/m2) | 55 (35.5) | 58 (47.5) | 113 (40.8) |
| Overweight (25.5–29.9 kg/m2) | 20 (12.9) | 4 (3.3) | 24 (8.7) |
| Obesity (30 kg/m2 or more) | 4 (2.6) | 1 (0.8) | 5 (1.8) |
| **Blood pressure** | | | |
| Normal (< 120/80 mmHg) | 37 (23.9) | 13 (10.7) | 50 (18.1) |
| Prehypertension (120-139/ 80–89 mmHg) | 105 (67.7) | 69 (56.6) | 174 (62.8) |
| Hypertension (>140/90 mmHg) | 13 (8.4) | 40 (32.8) | 53 (19.1) |
| **Glycated hemoglobin (HbA1c)\*** | | | |
| Normal (<5.7%) | 112 (74.5) | 86 (56.5) | 198 (73.3) |
| Prediabetes (5.7–6.5%) | 40 (25.5) | 32 (23.6) | 72 (26.7) |

*sum does not add up to the total due to missing

**Table 3. Change in food groups (servings per week) six months' post cafeteria intervention (n = 245).**

| Characteristic | Before Intervention (T1) | | After Intervention (T2) | | p-value* |
|---|---|---|---|---|---|
| | mean ± SD | median (IQR) | mean ± SD | median (IQR) | |
| Whole Grains | 0.51±1.84 | 0.00(0.00, 0.00) | 4.22±5.58 | 2.24(0.00, 7.00) | <0.001 |
| Refined Grains | 27.66±16.77 | 24.92(18.20, 33.25) | 21.23±11.34 | 19.85(13.62, 26.67) | <0.001 |
| Potato | 0.65±2.20 | 0.00(0.00, 0.00) | 0.52±1.55 | 0.00(0.00, 0.00) | 0.886 |
| Veg | 8.92±9.27 | 6.80(4.16, 10.72) | 10.26±7.93 | 9.05(5.48, 12.15) | <0.001 |
| Fruits | 1.92±4.15 | 0.00(0.00, 1.17) | 3.26±4.91 | 0.84(0.00, 4.79) | <0.001 |
| Fats Mono/Poly | 5.91±3.22 | 5.25(3.50, 7.82) | 8.73±4.11 | 8.13(5.83, 11.22) | <0.001 |
| Fat saturated | 0.28±1.14 | 0.00(0.00, 0.00) | 0.28±0.86 | 0.00(0.00, 0.00) | 0.629 |
| Lentil | 7.45±5.21 | 7.00(3.50, 10.50) | 8.45±8.57 | 7.00(3.50, 10.50) | 0.995 |
| Nuts | 0.37±2.09 | 0.00(0.00, 0.00) | 0.92±2.35 | 0.00(0.00, 0.02) | <0.001 |
| Red meat | 1.62±3.33 | 0.00(0.00, 1.75) | 1.71±3.31 | 0.00(0.00, 2.73) | 0.392 |
| White meat | 1.41±2.58 | 0.00(0.00, 1.75) | 1.12±2.14 | 0.00(0.00, 1.75) | 0.134 |
| Fish | 0.13±0.70 | 0.00(0.00, 0.00) | 0.38±1.98 | 0.00(0.00, 0.00) | 0.099 |
| Sugar Sweetened Beverages | 1.10±1.92 | 0.00(0.00, 2.10) | 1.12±2.24 | 0.00(0.00, 1.61) | 0.826 |

T1-before cafeteria intervention; T2-after cafeteria intervention; SD-Standard deviation; IQR-Interquartile range

* Wilcoxon signed rank test with continuity correction

**Table 4. Change in cardio-metabolic risk factors after cafeteria intervention (n = 257).**

| Variables | Mean (SD)Before Intervention | Mean (SD) After Intervention | Unadjusted change | | Adjusted Change | |
|---|---|---|---|---|---|---|
| | (T1) | (T2) | (T2-T1) (95% CI) * | p-value* | (T2-T1) (95% CI) ** | p-value |
| Systolic Blood pressure, mmHG | 116.30 (11.92) | 114.84 (11.83) | -1.71 (-2.84, -0.57) | 0.003 | -1.69 (-2.81, -0.57) | 0.003 |
| Diastolic Blood pressure, mmHg | 79.10 (9.19) | 78.22 (8.97) | -1.11 (-1.93, -0.29) | 0.008 | -1.08 (-1.89, -0.33) # | 0.01 |
| Glycated hemoglobin (HbA1c, %) | 5.56 (0.42) | 5.91(5.79) | 0.34 (-0.40, 1.09) | 0.368 | 0.32 (-0.36, 1.01) # | 0.370 |
| Fasting Blood Glucose, gm/dl | 91.19 (11.29) | 95.86(10.24) | 4.27 (2.89, 5.65) | <0.001 | 4.42 (3.11, 5.87) # | <0.001 |
| HDL cholesterol, gm/dl | 42.67 (12.35) | 42.16(14.44) | -0.26 (-2.15, 1.64)) | 0.790 | -0.34 (-2.09, 1.55) # | 0.720 |
| LDL cholesterol, gm/dl | 99.77 (28.85) | 93.21(25.83) | -6.69 (-9.54, -3.83) | <0.001 | -6.71 (-9.53, -3.89) | <0.001 |
| Total Cholesterol, gm/dl | 176.18 (38.91) | 182.86(103.20) | 6.34 (-6.62, 19.30) | 0.336 | 6.14 (-6.28, 19.17) # | 0.332 |
| Triglyceride, gm/dl | 137.14 (92.08) | 146.88(107.80) | 6.87 (-3.22, 16.97) | 0.181 | 7.15 (-3.15, 17.16) # | 0.158 |
| Body Mass Index, kg/m$^2$ | 25.47 (4.40) | 25.47(3.55) | -0.07 (-0.51, 0.38) | 0.774 | -0.02 (-0.44, 0.42) # | 0.924 |

T1-before cafeteria intervention; T2-after cafeteria intervention; SD-Standard deviation; CI: confidence interval; HDL: High Density Lipoprotein; LDL: Low Density Lipoprotein

* Paired t-test

** Linear mixed-effect model adjusting for age, gender, ethnicity, education, marital, and religion and including participant unique identity as a random effect

# CI generated using bootstrapping where the residuals of the model were not normally distributed

dietary intake and cardiometabolic risk factors. Specifically, after six months of cafeteria intervention, there was a significant increase in the intake of whole grains, fruits, nuts, mono/polyunsaturated oil, and fish, whereas the intake of refined grains decreased. There was a slight improvement in systolic and diastolic blood pressure and low-density lipoprotein. However, there was an increase in mean fasting blood sugar level.

In our study, we found an increase in the consumption of healthy foods and a decrease in the consumption of unhealthy foods. Systematic reviews have shown that strategies promoting healthy foods in the workplace, particularly environmental changes, have improved the intake of healthy foods [15, 38–41]. Our intervention used a multi-component environmental approach to promote healthy choices. A number of important factors led to the positive changes–increased availability of healthy food options, advertisement of healthy foods, and engagement of employees and cafeteria managers in planning and monitoring [42].

We observed a decrease in blood pressure six months after the cafeteria intervention in the workplace by adding whole grains, fruits, and vegetables and reducing refined grains and sugar-sweetened beverages. A large reduction in blood pressure (13.2 mmHg SBP and 14.9 mmHg DBP) was observed after introducing a Mediterranean diet in a workplace cafeteria over a year [24]. Another intervention study that implemented a low-fat vegan diet for 22 weeks resulted in no change in blood pressure, but it rose in the control group by approximately 5 mmHg at the same time [43]. The DASH (Dietary Approaches to Stop Hypertension) trial demonstrated a diet that emphasizes fruits, vegetables, and low-fat dairy products, including whole grains, poultry, fish, and nuts, while limiting red meat, sweets, and sugar-containing beverages. Total and saturated fat lowered blood pressure substantially in people with and without hypertension [44–46]. Our study has shown an increase in the intake of whole grains, fruits, nuts, mono/polyunsaturated fat, and fish and a reduction in the intake of refined grains. This could mediate blood pressure reduction and decrease the LDL level in the blood.

It is unclear why fasting blood glucose increased during the cafeteria intervention period. Fasting blood glucose levels might have been affected by a recent diet. There was no change in HbA1c during the cafeteria intervention period. BMI did not decrease over the intervention year. However, mean BMI over the past decade has been increasing worldwide [47]; the

observed lack of increase in BMI may be due to a beneficial effect of the intervention. A study of health appraisal found no change in BMI over the intervention but increased BMI in the control group that did not receive any appraisal over 2 years [23]. It is argued that the impact achieved in a wellness program should be compared to the likely natural decline in the health status of the population that results from aging and the natural progression of risk in the absence of intervention [4].

We did not find a significant change in total cholesterol, triglycerides, or HDL. However, mean LDL decreased significantly by 7 mg/dL six months after the start of the cafeteria intervention. Worksite interventions have found a similar decrease in LDL through a low-fat plant-based diet [21]. A factory-based intervention of a low-calorie, increased fiber, and unsaturated fat diet found significantly lower mean serum cholesterol of nearly 10%, increased HDL, and decreased LDL after two years [48]. Finally, an intervention with a low-fat vegan diet for 22 weeks led to a decrease in total and low-density lipoprotein that was not statistically significant, while HDL significantly decreased in the intervention group [43].

The environmental-level changes to the cafeteria were modifications to offer healthy food; there were no other changes in the worksite's overall health policies. Therefore, employees had the choice of whether or not to take the healthier options introduced by the cafeteria intervention. Cafeteria interventions that mandate the employees to take only healthy foods may improve greater behavior change [49]. However, environmental-level changes may not be sufficient to bring the behavior changes that impact health. According to the Social Cognitive Theory, a person's behavior is caused by an interaction of cognitive, affective, and environmental influences [50]. Healthy environments may keep low-risk employees at low risk and improve health outcomes for higher-risk employees [51].

The study has several strengths. This is the first study to report a positive effect of worksite-based environmental level changes on cardiometabolic disorders in Nepal. The employee participation rate was high. About 95% of those invited participants in the screening and 85% of eligible participants were enrolled, and 70% were retained at the end of the first year. The major reason for discontinuation was resignation or work leave for further study. This shows that it is feasible to implement a worksite CVD intervention in this setting. As mentioned in the published study protocol, the intervention was designed using a participatory approach, where catering and workplace stakeholders were involved in the study design and implementation of the interventions in the individual workplaces [25]. Participatory and theory-based approaches to workplace health promotion have been recommended to ensure the effectiveness of nutrition in workplace health promotion [52]. Finally, we used objective measurements to assess cardiometabolic risk factors, minimizing the information bias. The before-after design controls for confounding by all time-invariant risk factors.

Several limitations should be considered when interpreting these results. First, there is a possibility of regression to the mean bias [53], especially in the results six months post health risk appraisal, due to random within-person variation. Second, the study may have limited generalizability because the results are employees of one hospital in central Nepal. However, the results highlight the need for health assessments and more intensive health promotion efforts in similar worksites throughout Nepal, suitably contextualized and evaluated. Further, the socio-demographic characteristics of the participants who completed the study were mostly similar to those of those who dropped out (S2 Table. Comparison of characteristics of participants who completed the study and who dropped out of the study). However, those who dropped out were, in general, older than those who completed the study and had higher chances of being overweight and pre-diabetic. This might have limited the study's generalizability. However, the age, overweight prevalence, and prediabetes prevalence of drop-outs were similar between the intervention and control groups. Third, there was no comparison group in this study because it

was not feasible to intervene in some of the four worksite cafeterias and not others since employees were allowed to go to any cafeteria of their choice. The participants' food intake and health outcomes were compared before and after the intervention, so we could not control for confounding by time and other variables that change with time. Fourth, social desirability reporting bias cannot be ruled out either, as employees with higher nutrition knowledge may have overestimated their intake of healthy foods in the dietary recalls.

## Conclusion

Overall, consumption of healthy foods increased, and blood pressure and LDL had clinically irrelevant reductions six months after an environmental-level health intervention incorporating access to healthy foods in the cafeteria. There was no change in HbA1c and BMI. This is slightly beneficial compared to the expected increase in these indicators as a natural risk progression without intervention. Worksite can offer a unique intervention environment to improve the health of its employees by reducing risk factors and promoting healthy behaviors. However, more studies with modified interventions are needed to study its effect on cardiometabolic risk factors.

## Supporting information

**S1 Table. List of food added.**
(DOCX)

**S2 Table. Comparison of characteristics of participants who completed the study and who dropped out of the study.**
(DOCX)

**S1 Checklist. Reporting checklist for randomised trial.**
(DOCX)

**S1 File.**
(DOCX)

**S2 File.**
(DOCX)

**S3 File.**
(DOCX)

**S4 File.**
(JPG)

**S5 File.**
(DOCX)

**S6 File.**
(DOCX)

**S1 Dataset.**
(CSV)

## Acknowledgments

We acknowledge Dhulikhel Hospital- Kathmandu University Hospital for their permission and cooperation in conducting this study. We appreciate the contribution of all study participants.

## Author Contributions

**Conceptualization:** Archana Shrestha, Dipesh Tamrakar, Bhawana Ghinanju, Vasanti Malik, Josiemer Mattei, Donna Spiegelman.

**Data curation:** Archana Shrestha, Deepa Shrestha, Parashar Khadka, Bikram Adhikari, Jayana Shrestha, Suruchi Waiwa, Prajjwal Pyakurel, Niroj Bhandari, Biraj Man Karmacharya, Akina Shrestha, Rajeev Shrestha, Rajendra Dev Bhatta, Vasanti Malik, Josiemer Mattei, Donna Spiegelman.

**Formal analysis:** Archana Shrestha, Niroj Bhandari, Josiemer Mattei, Donna Spiegelman.

**Funding acquisition:** Archana Shrestha, Donna Spiegelman.

**Investigation:** Archana Shrestha.

**Methodology:** Archana Shrestha.

**Project administration:** Archana Shrestha, Jayana Shrestha, Suruchi Waiwa.

**Resources:** Archana Shrestha.

**Software:** Archana Shrestha.

**Supervision:** Archana Shrestha.

**Validation:** Archana Shrestha.

**Visualization:** Archana Shrestha.

**Writing – original draft:** Archana Shrestha, Bikram Adhikari, Niroj Bhandari.

**Writing – review & editing:** Archana Shrestha, Dipesh Tamrakar, Bhawana Ghinanju, Deepa Shrestha, Parashar Khadka, Bikram Adhikari, Jayana Shrestha, Suruchi Waiwa, Prajjwal Pyakurel, Niroj Bhandari, Biraj Man Karmacharya, Akina Shrestha, Rajeev Shrestha, Rajendra Dev Bhatta, Vasanti Malik, Josiemer Mattei, Donna Spiegelman.

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
