## [Decision Letter · Decision Letter 0]

29 Sep 2023

PONE-D-23-17601Effects of a dietary intervention on cardiometabolic risk and food consumption in a workplacePLOS ONE

Dear Dr. Shrestha,

Thank you for submitting your manuscript to PLOS ONE. After careful consideration, we feel that it has merit but does not fully meet PLOS ONE’s publication criteria as it currently stands. Therefore, we invite you to submit a revised version of the manuscript that addresses the points raised during the review process.

We look forward to receiving your revised manuscript.

Kind regards,

Victor Manuel Mendoza-Nuñez, PhD

Academic Editor

PLOS ONE

Journal Requirements:

- https://cora.ucc.ie/server/api/core/bitstreams/0557a56a-56e1-439b-a28b-afab69ac1d43/content

- https://www.cambridge.org/core/journals/public-health-nutrition/article/health-impact-of-mediterranean-diets-in-food-at-work/DDE6DEC9914DEEFB155EF0979E4D636F

- https://bmccardiovascdisord.biomedcentral.com/articles/10.1186/s12872-019-1025-3

- https://bmcpublichealth.biomedcentral.com/articles/10.1186/s12889-016-2828-0

In your revision ensure you cite all your sources (including your own works), and quote or rephrase any duplicated text outside the methods section. Further consideration is dependent on these concerns being addressed.

"This was a funded study

 Initials of the authors who received each award: AS and DS

 Grant number awarded to each author: DP1ES025459

 Full name of the funder: National institute of health (NIH)

 URL: https://www.nih.gov/

 No sponsors did not play any role."

5. Please include a caption for figure 1.

**Additional Editor Comments:**

REVIEWER 1

The report addresses an interesting topic applied in a challenging context. Getting the involvement of a commercial instance that offers food to a large group of people is a merit that should be recognized.

Overall, the work is well presented. The outcome variables address risk factors relevant to the prevention and management of different cardiometabolic conditions.

Although the design is, within the group of experimental designs, the one with the least explanatory power, the context of the development of the study justifies it.

In the opinion of this reviewer the main limitations of the study are two:

1) The final study dropout rate among those eligible is almost 30% (n=109). It would be useful to compare the sociodemographic and cardiometabolic profiles of those who completed the study and those who dropped out in order to estimate or rule out a possible selection bias.

2) At first glance, Tables 1 and 2 show notable differences between the sexes. Analysis based solely on bivariate estimates for related samples is limited in light of the observed baseline differences between men and women. It seems necessary to use a multivariate approach adjusting for sociodemographic variables or, at least by sex, and for those variables in which there are significant cardiometabolic differences in baseline values.

Form and typographical issues:

1) The number of participants who rejected the first call do not match, in the text it says 35 and in Figure 1 it says 33.

2) It would be convenient to add a column of p-values in Tables 1 and 2 to appreciate the significance of the differences between sexes.

3) It is unnecessary to repeat in the text the data presented in Tables 3 and 4. The reader can clearly identify them in the text. It is only necessary to mention in which variables and in which direction the changes were significant.

4) To explore other hypotheses about the unexpected result of the increase in fasting glucose. It is possible that the multivariate analysis approach could provide some explanation.

5) It is unnecessary to repeat the expression cafeteria intervention. "intervention" is enough.

REVIEWER 2

The authors mention that after six months of cafeteria intervention to facilitate intake of healthy foods and discourage the increase in unhealthy foods, they obtained statistically significant changes in systolic and diastolic blood pressure, as well as a decrease in low-density lipoproteins (LDL). However, these differences do not show any clinical significance in the participants, so no benefit is observed in them. On the other hand, an increase in blood glucose levels was observed and the glycated hemoglobin, body mass index, triglycerides and total cholesterol of the participants remained unchanged. Therefore, despite a change in the consumption of healthy foods, there is no impact on the participants, hence the results of this article are negative since a decrease is not observed in the markers of cardiometabolic risk and the consumption of food.

Likewise, when reviewing the baseline data of their work, it can be seen that 73.3% of the participants showed a glycated hemoglobin value within normal values, approximately 90% of the participants have a low or normal BMI and 60% performed a high physical activity according to the calculated mets, so better results would hardly be obtained.

Therefore, the authors are recommended that the results be modified towards negative results of their intervention proposal or only maintenance among the participants, proposing another or improving the current one, where healthy environments can keep low-risk employees in a low level and perhaps improve health outcomes for higher-risk employees by working with this group of participants.

**Comments academic editor**

I agree with reviewer 2, the effect of the intervention is clinically irrelevant, therefore, the authors are recommended to give a negative approach to the intervention, analyzing the reasons for not having observed the expected effect. It is also necessary for the authors to include a reasoned proposal (modifications to the intervention carried out) for future studies.

Reviewers' comments:

Reviewer's Responses to Questions

**Comments to the Author**

1. Is the manuscript technically sound, and do the data support the conclusions?

Reviewer #1: Partly

Reviewer #2: Partly

2. Has the statistical analysis been performed appropriately and rigorously? 

Reviewer #1: No

Reviewer #2: Yes

3. Have the authors made all data underlying the findings in their manuscript fully available?

Reviewer #1: Yes

Reviewer #2: Yes

4. Is the manuscript presented in an intelligible fashion and written in standard English?

Reviewer #1: Yes

Reviewer #2: Yes

5. Review Comments to the Author

Reviewer #1: The report addresses an interesting topic applied in a challenging context. Getting the involvement of a commercial instance that offers food to a large group of people is a merit that should be recognized.

Overall, the work is well presented. The outcome variables address risk factors relevant to the prevention and management of different cardiometabolic conditions.

Although the design is, within the group of experimental designs, the one with the least explanatory power, the context of the development of the study justifies it.

In the opinion of this reviewer the main limitations of the study are two:

1) The final study dropout rate among those eligible is almost 30% (n=109). It would be useful to compare the sociodemographic and cardiometabolic profiles of those who completed the study and those who dropped out in order to estimate or rule out a possible selection bias.

2) At first glance, Tables 1 and 2 show notable differences between the sexes. Analysis based solely on bivariate estimates for related samples is limited in light of the observed baseline differences between men and women. It seems necessary to use a multivariate approach adjusting for sociodemographic variables or, at least by sex, and for those variables in which there are significant cardiometabolic differences in baseline values.

Form and typographical issues:

1) The number of participants who rejected the first call do not match, in the text it says 35 and in Figure 1 it says 33.

2) It would be convenient to add a column of p-values in Tables 1 and 2 to appreciate the significance of the differences between sexes.

3) It is unnecessary to repeat in the text the data presented in Tables 3 and 4. The reader can clearly identify them in the text. It is only necessary to mention in which variables and in which direction the changes were significant.

4) To explore other hypotheses about the unexpected result of the increase in fasting glucose. It is possible that the multivariate analysis approach could provide some explanation.

5) It is unnecessary to repeat the expression cafeteria intervention. "intervention" is enough.

Reviewer #2: The authors mention that after six months of cafeteria intervention to facilitate intake of healthy foods and discourage the increase in unhealthy foods, they obtained statistically significant changes in systolic and diastolic blood pressure, as well as a decrease in low-density lipoproteins (LDL). However, these differences do not show any clinical significance in the participants, so no benefit is observed in them. On the other hand, an increase in blood glucose levels was observed and the glycated hemoglobin, body mass index, triglycerides and total cholesterol of the participants remained unchanged. Therefore, despite a change in the consumption of healthy foods, there is no impact on the participants, hence the results of this article are negative since a decrease is not observed in the markers of cardiometabolic risk and the consumption of food.

Likewise, when reviewing the baseline data of their work, it can be seen that 73.3% of the participants showed a glycated hemoglobin value within normal values, approximately 90% of the participants have a low or normal BMI and 60% performed a high physical activity according to the calculated mets, so better results would hardly be obtained.

Therefore, the authors are recommended that the results be modified towards negative results of their intervention proposal or only maintenance among the participants, proposing another or improving the current one, where healthy environments can keep low-risk employees in a low level and perhaps improve health outcomes for higher-risk employees by working with this group of participants.

6. PLOS authors have the option to publish the peer review history of their article (what does this mean?). If published, this will include your full peer review and any attached files.

Reviewer #1: No

Reviewer #2: No

---

## [Author Response · Author response to Decision Letter 0]

13 Nov 2023

Nov 10, 2023

To,

The Editorial Team, 

PloS One Journal

Dear Reviewers,

It is with enthusiasm that I resubmit a revised version of the manuscript titled “Effects of a Dietary Intervention on Cardiometabolic Risk and Food Consumption in a Workplace”.This study was funded by the National Institute of Health Director’s Pioneer award ( 5DP1ES025459). The funders had no role in the study design, data collection, analysis, decision to publish, or preparation of the manuscript.

Thank you for allowing us to revise and resubmit this manuscript. I appreciate the time and effort provided by the reviewer. I have incorporated the suggested changes into the manuscript to the best of my ability. The manuscript has certainly benefited from these insightful suggestions. I look forward to working with you and the reviewer to move this manuscript closer to publication in the Journal of PLOS ONE.

Append to this letter is our response to the comments raised by the reviewer and the editor. As you will notice, we agreed with all the comments raised by the reviewers. Accordingly, we have uploaded a copy of the original manuscript marked with all the changes in track change made during the revision process, as well as the revised clean version.

Thank you, 

Archana Shrestha,

Associate Professor

Department of Public Health

Kathmandu University School of Medical Sciences

Journal Requirements:

Thank you for the comments. We have reviewed the requirements and made changes in font size in the level 1, 2, and 3 headings with the font sizes 18, 14 and 16. 

- https://cora.ucc.ie/server/api/core/bitstreams/0557a56a-56e1-439b-a28b-afab69ac1d43/content

-https://www.cambridge.org/core/journals/public-health-nutrition/article/health-impact-of-mediterranean-diets-in-food-at-work/DDE6DEC9914DEEFB155EF0979E4D636F

- https://bmccardiovascdisord.biomedcentral.com/articles/10.1186/s12872-019-1025-3

- https://bmcpublichealth.biomedcentral.com/articles/10.1186/s12889-016-2828-0

In your revision ensure you cite all your sources (including your own works), and quote or rephrase any duplicated text outside the methods section. Further consideration is dependent on these concerns being addressed.

The overlapping statements have been paraphrased and indicated in track-change mode in the article. The paraphrased sentences are as follows

In the Introduction section: In contrast to focusing on a small number of population, social-ecological approaches that address social and environmental factors to prevent cardiometabolic risk reach out to a larger sector of the population.

In the methods section: We collected blood samples using evacuated blood collection tubes.

In the discussion section: Systolic blood pressure was reduced by 13 mmHg and diastolic blood pressure was reduced by 15 mmHg with a Mediterranean diet-based workplace intervention. 

We have updated the information in the funding section of the manuscript. 

"This was a funded study Initials of the authors who received each award: AS and DS

 Grant number awarded to each author: 5DP1ES025459 Full name of the funder: National institute of health (NIH) URL: https://www.nih.gov/

 No sponsors did not play any role."

Please state what role the funders took in the study. If the funders had no role, please state: 

"The funders had no role in study design, data collection and analysis, decision to publish, or preparation of the manuscript." If this statement is not correct you must amend it as needed. 

We have added this information to the cover letter. 

5. Please include a caption for figure 1.

Response:

Caption for figure 1 is included in the manuscript as follows;

Figure 1. Flow diagram study participants of the Nepal Pioneer Worksite Intervention Study

Response:

We have included captions for Supporting Information files at the end of the manuscript as follows; 

Supporting file information:

S1 - Table S1. List of food added

S2 - Table S2. Comparison of characteristics of participants who completed the study and who dropped out. 

S3 - Approval from Kathmandu University School of Medical Sciences(KUSMS) IRC

S4 - CONSORT Checklist

S5 - Harvard IRB initial study approval

S6 - Details on the study protocol 

S7 - Modification application to KUSMS IRC

S8 - Modification approval from KUSMS IRC

S9 - Modification approval from Harvard IRB

Additional Editor Comments:

REVIEWER 1

The report addresses an interesting topic applied in a challenging context. Getting the involvement of a commercial instance that offers food to a large group of people is a merit that should be recognized. Overall, the work is well presented. The outcome variables address risk factors relevant to the prevention and management of different cardiometabolic conditions.

Although the design is, within the group of experimental designs, the one with the least explanatory power, the context of the development of the study justifies it.

In the opinion of this reviewer the main limitations of the study are two:

1) The final study dropout rate among those eligible is almost 30% (n=109). It would be useful to compare the sociodemographic and cardiometabolic profiles of those who completed the study and those who dropped out in order to estimate or rule out a possible selection bias.

Response:

Thank you for the comments. We compared the sociodemographic and cardiometabolic profiles of the 257 participants who completed the study and the 56 participants who dropped out. The tables have been submitted as supplemental files and we have added the information in the discussion section as follows. 

“The socio-demographic characteristics of the participants who completed the study were mostly similar to those of those who dropped out (Supplementary Table X). However, those who dropped out were in general older than those who completed the study, and had higher chances of being overweight and pre-diabetic. This might have limited the study’s generalizability.”

2) At first glance, Tables 1 and 2 show notable differences between the sexes. Analysis based solely on bivariate estimates for related samples is limited in light of the observed baseline differences between men and women. It seems necessary to use a multivariate approach adjusting for sociodemographic variables or, at least by sex, and for those variables in which there are significant cardiometabolic differences in baseline values.

We have updated the results in Table 4, adjusted for age, gender, education, ethnicity and religion. This statistical analysis is also updated in the methods section as follows:

In the method section: 

We employed a mixed-effects linear regression model with long data, wherein participant ID served as the random effect variable, and time or wave functioned as the independent variable, adjusted for socio-demographic variables such as age, gender, education, ethnicity, and religion

Table 4. Change in cardio-metabolic risk factors after cafeteria intervention (n=257)

 Cardio-metabolic risk factors Before Intervention 

mean (SD) 

(T1) After Intervention

mean (SD 

(T2) Crude Difference 

mean 

(95% CI)

(T2-T1) p-value Adjusted Mean Difference (T2-T1)

(95% CI) p-value

Systolic Blood Pressure, mmHg 116.3 (11.9) 114.8 (11.8) -1.7 ( -2.8,-0.6) 0.003 -1.7 (-2.8, -0.6 0.003

Diastolic Blood Pressure, mmHg 79.1 (9.2) 78.2 (8.9) -0.1 ( -1.9,-0.3) 0.008 -1.1(-1.9 , -0.3) 0.01

Fasting blood sugar, mg/dL 91.2 (11.3) 95.7 (10.2) 4.3 (2.9,5.6) <0.001 4.4(3.0, 5.8) <0.001

Glycated hemoglobin, HbA1c 5.5 (0.4) 5.5 (0.4) 0.1 (0.1,0.2) 0.06 0.3(-0.4, 1.0) 0.37

Triglyceride, mg/dL 137.2 (92.1) 147.2 (107.1) 6.8 (-3.2, 16.9) 0.18 7.1(-2.8, 17.1) 0.158

High Density Lipoprotein, mg/dL 42.7(12.4) 42.1 (14.3) -0.3 (-2.1,1.6) 0.79 -0.3(-2.2, 1.5) 0.72

Low Density Lipoprotein, mg/dL 99.7 (28.5) 92.9 (25.7) -6.7 (-9.5, -3.8) <0.001 -6.7(-9.5, -3.9) <0.001

Total Cholesterol, mg/dL 176.2 (38.9) 182.4 (102.1) 6.3 (-3.2,16.9) 0.33 6.1(-6.3, 18.5) 0.332

BMI, kg/m2 25.3 (3.4) 25.3 (3.4) 0.01 (0.03,0.05) 0.90 -0.1(-0.5, 0.4) 0.924

Adjusted for age, gender, education, ethnicity, and religion. 

T1-before cafeteria intervention, T2-after cafeteria intervention, SD-Standard deviation CI: Confidence Interval

In the methods section: 

Further, we utilized linear mixed model (estimated using REML and nloptwrap optimizer) to predict cardiometabolic risk factors, including HbA1c (%mg/dl), systolic and diastolic blood pressure(mmHg), and lipids(blood pressure (mg/dL) variables with intervention (pre/post) adjusting for age, gender, ethnicity, education, marital, and religion and including participant unique ID as a random effect. 

Form and typographical issues:

1) The number of participants who rejected the first call do not match, in the text it says 35, and in Figure 1 it says 33.

Thank you for noticing this. This has been corrected in the manuscript figure 1 as Refused (not interested) = 35 (4.2%)

2) It would be convenient to add a column of p-values in Tables 1 and 2 to appreciate the significance of the differences between sexes.

We have added the p-values in Tables 1 and 2. 

3) It is unnecessary to repeat in the text the data presented in Tables 3 and 4. The reader can clearly identify them in the text. It is only necessary to mention in which variables and in which direction the changes were significant.

This has been addressed in the track changes of the manuscript as: 

Six months post cafeteria intervention, the mean consumption of whole grains increased by 3.7 servings per week ( 95% CI: 3.0,4.5; p<0.001); mono/polyunsaturated fat increased by 2.9 servings per week (95% CI:2.3,3.5; p<0.001); fruits increased by 1.5 servings per week (95% CI: 0.8,2.3; p<0.001); and nuts increased by 0.5 servings per week (95% CI:0.2,0.9;p<0.001). The mean consumption of refined grains decreased by 7.3 servings per week (95% CI: -9.7, -4.9; p<0.001). 

The CVD risk factors and biomarkers before and after intervention are presented in Table 4. 

The mean systolic blood pressure and diastolic blood pressure decreased significantly by 1.7 mm of Hg (95% CI: -2.8,-0.6;p=0.003) and 1.1 mm of Hg (95% CI: -1.9, -0.3;p=0.01) respectively after cafeteria intervention after adjusting for age, gender, education, ethnicity, and religion. Mean LDL decreased significantly by 6.7 mg/dL (95% CI:-9.5, -3.9; p<0.001) after the cafeteria intervention after adjusting for age, gender, education, ethnicity, and religion. The mean fasting blood sugar increased by 4.3 mg/dL(95% CI:3.0, 5.8; p<0.001) during the intervention period in the multivariate model.

4) To explore other hypotheses about the unexpected result of the increase in fasting glucose. It is possible that the multivariate analysis approach could provide some explanation.

Thank you for the comment. We have added multivariate analysis and found that the results were not different. The Multivariate analysis results are added in Table 4. 

5) It is unnecessary to repeat the expression cafeteria intervention. "intervention" is enough.

We have consistently used ‘intervention’ in the manuscript now.

REVIEWER 2

The authors mention that after six months of cafeteria intervention to facilitate intake of healthy foods and discourage the increase in unhealthy foods, they obtained statistically significant changes in systolic and diastolic blood pressure, as well as a decrease in low-density lipoproteins (LDL). However, these differences do not show any clinical significance in the participants, so no benefit is observed in them. On the other hand, an increase in blood glucose levels was observed and the glycated hemoglobin, body mass index, triglycerides, and total cholesterol of the participants remained unchanged. Therefore, despite a change in the consumption of healthy foods, there is no impact on the participants, hence the results of this article are negative since a decrease is not observed in the markers of cardiometabolic risk and the consumption of food.

Likewise, when reviewing the baseline data of their work, it can be seen that 73.3% of the participants showed a glycated hemoglobin value within normal values, approximately 90% of the participants have a low or normal BMI and 60% performed a high physical activity according to the calculated mets, so better results would hardly be obtained.

Therefore, the authors are recommended that the results be modified towards negative results of their intervention proposal or only maintenance among the participants, proposing another or improving the current one, where healthy environments can keep low-risk employees in a low level and perhaps improve health outcomes for higher-risk employees by working with this group of participants.

Comments academic editor

I agree with reviewer 2, the effect of the intervention is clinically irrelevant, therefore, the authors are recommended to give a negative approach to the intervention, analyzing the reasons for not having observed the expected effect. It is also necessary for the authors to include a reasoned proposal (modifications to the intervention carried out) for future studies.

We agree with the comments. We have made the changes in the interpretation of the study discussion and conclusion section as follows. 

In the discussion section: 

There was a slight improvement in systolic and diastolic blood pressure and low-density lipoprotein but the improvement was not clinically significant.

However, the change was small and clinically irrelevant, which may be due to short study duration, and having a large proportion of the study participants to have normal BMI and being physically active at baseline.

These are the changes made in the conclusion section:

Overall, consumption of healthy foods increased, and blood pressure and LDL had clinically irrelevant reductions six months after an environmental-level health intervention incorporating access to healthy foods in the cafeteria. There was no change in HbA1c and BMI. This is slightly beneficial compared to the expected increase in these indicators as a natural progression of risk in the absence of intervention. Worksite can offer a unique intervention environment to improve the health of its employees by reducing risk factors and promoting healthy behaviors but more studies with modified interventions are needed to study its effect on cardiometabolic risk factors. 

Supplementary Table S2. Comparison of characteristics of participants who completed the study and who dropped out of the study. 

Characteristics Completed the study

(n = 257)

n (%) Drop out from the study

(n = 56)

n (%) P-value

Age, years Mean (SD) 31.1±7.6 37.9±8.6 <0.0001

Gender 

Male 105(40.9) 16(28.6) 0.087

Female 152(59.1) 40(71.4) 

Ethnicity 

 Brahmin/Chhetri 93(36.2) 21(37.5) 0.539

 Newar 122(47.5) 30(53.6) 

 Tamang 29(11.3) 3(5.4) 

 Others 13(5.1) 2(3.6) 

Marital Status 

 Married 175(68.1) 45(80.4) 0.003

 Not Married 82(31.9) 11(19.6) 

Religion 

 Hindu 229(89.1) 52(92.9) 0.816

 Not Hindu 28(10.9) 4(7.1) 

Education 

 Less than high school 73(28.4) 15(26.8) 0.807

 High School or more 184(71.6) 42(73.2) 

 Years of education, Mean (SD) 12.7±3.8 

Smoking 

 Never 198(77.0) 42(75.0) 0.805

 Former 17(6.6) 3(5.4) 

 Current 42(16.3) 11(19.6) 

Alcohol intake (drinks per day) 

 Never 121(47.1) 26(47.3) 0.304

 1 or less drinks per day 112(43.5) 24(43.6) 

 1- 2 drinks per day 11(4.3) 0 

 3 or more drinks per day 13(5.1) 5(9.1) 

Physical activity 

 Low (<600 METmin/ week) 93(41.5) 13(27.7) 0.077

 High (600 or more METmin/week) 131(58.5) 34(72.3) 

Body Mass Index 

 Underweight (18.5 kg/m2 or less) 4(1.6) 1(1.8) 0.008

 Normal (18.6 – 24.9 kg/m2) 139(54.9) 17(30.4) 

 Overweight (25.0 – 29.9 kg/m2) 94(36.6) 34(60.7) 

 Obesity (30 kg/m2 or more) 20(7.8) 4(7.1) 

Blood pressure 

 Normal (< 120/80 mmHg) 76(29.6) 56(100.0) -

 Prehypertension (120-139/ 80-89 mmHg) 147(57.2) 0 

 Hypertension (>140/90 mmHg) 34(13.2) 0 

Glycated hemoglobin (HbA1c)* 

 Normal (<5.7%) 188(80.3) 23(41.9) <0.001

 Prediabetes (5.7- 6.5%) 46(19.7) 33(58.9)

---

## [Decision Letter · Decision Letter 1]

5 Jan 2024

PONE-D-23-17601R1Effects of a dietary intervention on cardiometabolic risk and food consumption in a workplacePLOS ONE

Dear Dr. Shrestha,

Thank you for submitting your manuscript to PLOS ONE. After careful consideration, we feel that it has merit but does not fully meet PLOS ONE’s publication criteria as it currently stands. Therefore, we invite you to submit a revised version of the manuscript that addresses the points raised during the review process.

We look forward to receiving your revised manuscript.

Kind regards,

Victor Manuel Mendoza-Nuñez, PhD

Academic Editor

PLOS ONE

Journal Requirements:

Reviewers' comments:

Reviewer's Responses to Questions

**Comments to the Author**

1. If the authors have adequately addressed your comments raised in a previous round of review and you feel that this manuscript is now acceptable for publication, you may indicate that here to bypass the “Comments to the Author” section, enter your conflict of interest statement in the “Confidential to Editor” section, and submit your "Accept" recommendation.

Reviewer #1: All comments have been addressed

Reviewer #2: All comments have been addressed

Reviewer #3: (No Response)

2. Is the manuscript technically sound, and do the data support the conclusions?

Reviewer #1: Yes

Reviewer #2: Yes

Reviewer #3: Yes

3. Has the statistical analysis been performed appropriately and rigorously? 

Reviewer #1: Yes

Reviewer #2: Yes

Reviewer #3: Yes

4. Have the authors made all data underlying the findings in their manuscript fully available?

Reviewer #1: Yes

Reviewer #2: Yes

Reviewer #3: No

5. Is the manuscript presented in an intelligible fashion and written in standard English?

Reviewer #1: Yes

Reviewer #2: Yes

Reviewer #3: Yes

6. Review Comments to the Author

Reviewer #1: All my comments were sufficiently addressed. It may be worth emphasizing the inherent limitation imposed by the quasi-experimental design employed, which not only limits the external validity of the study, but also its external validity.

Reviewer #2: The authors have considered the recommended suggestions according to the results obtained in their manuscript.

Reviewer #3: The manuscript addresses an interesting topic. The employed statistical methods are rather sound.

To ensure the reliability of the results and their reproducibility the following aspects should be further investigated:

1. Any parametric tests rely on quite strong assumptions. This is for sure true for the paired t-test considered in this work. Please, check and provide evidence that all the assumptions are met.

2. I really like the idea of using linear mixed models. More details about the random effects distributions and the estimation method should be given; please, check for the Gaussian distribution of the random effects, or for whatever distribution you assumed for the random terms. Moreover, Gauss-Markov assumption should be further checked. A detailed residual analysis is fundamental to appreciate the reliability of the results.

7. PLOS authors have the option to publish the peer review history of their article (what does this mean?). If published, this will include your full peer review and any attached files.

Reviewer #1: No

Reviewer #2: No

Reviewer #3: No

---

## [Author Response · Author response to Decision Letter 1]

26 Feb 2024

February 18, 2024

To,

The Editorial Team, 

PloS One Journal

Dear Reviewers,

I am enthusiastic about resubmitting a revised version of the manuscript titled “Effects of a Dietary Intervention on Cardiometabolic Risk and Food Consumption in a Workplace.” This study was funded by the National Institute of Health Director’s Pioneer award (5DP1ES025459). The funders had no role in the study design, data collection, analysis, publication decision, or manuscript preparation.

Thank you for allowing us to revise and resubmit this manuscript. I appreciate the time and effort provided by the reviewer. I have incorporated the suggested changes into the manuscript to the best of my ability. The manuscript has undoubtedly benefited from these insightful suggestions.

I look forward to working with you and the reviewer to move this manuscript closer to publication in the Journal of PLOS ONE.

Append to this letter is our response to the comments raised by the reviewer and the editor. As you will notice, we agreed with all the comments raised by the reviewers. Accordingly, we have uploaded a copy of the original manuscript marked with all the changes in track change made during the revision process and the revised clean version.

Thank you, 

Archana Shrestha,

Associate Professor

Department of Public Health

Kathmandu University School of Medical Sciences

Response to Reviewer’s comments

1. Any parametric tests rely on quite strong assumptions. This is for sure true for the paired t-test considered in this work. Please check and provide evidence that all the assumptions are met.

Ans: Thank you for your feedback. I checked the assumptions and discovered some of the assumptions were met. Therefore, we decided to apply a non-parametric test. The results are not different. We have revised the methods and results section accordingly, as below.

Methods:

We applied a paired test to compare normally distributed variables like systolic blood pressure, diastolic blood pressure, BMI, HDL, HbA1c, and fasting blood glucose before and after intervention. We used paired Wilcoxon tests to compare non-normally distributed variables such as food groups (servings per week) before and after intervention. We compared the change in food group (servings per week), HbA1c (mg/dl), systolic blood pressure(mmHg), and lipids before and after cafeteria intervention and six months after using the paired t-test. 

Results: We have revised Table 3 based on the updated analysis

2. I like the idea of using linear mixed models. More details about the random effects distributions and the estimation method should be given; please check for the Gaussian distribution of the random effects or for whatever distribution you assumed for the random terms. Moreover, the Gauss-Markov assumption should be further checked. A detailed residual analysis is fundamental to appreciate the reliability of the results.

Thank you for your feedback. We have revised the method and result sections as per your suggestion as follows:

Methods: We utilized a linear mixed-effect model to determine and assess the difference in cardiometabolic risk factors, including HbA1c (%mg/dl), systolic and diastolic blood pressure(mmHg), and lipids (blood pressure (mg/dL) before and after the interventions, adjusting for age, gender, ethnicity, education, marital, and religion and participant unique identity as a random effect. We checked the assumptions of the random effects, which were found to follow a Gaussian distribution. We checked the Gauss-Markov assumption and performed a detailed residual analysis to ensure the reliability of the results. We used bootstrapping to estimate the confidence interval of the linear mixed-effect model parameters if the residuals were not normally distributed. We drew 1000 bootstrap samples from the original data set with replacement and computed the 95% confidence intervals using the “boot” package. We used R version 4.3.1 to perform pre-analytical processing and statistical analysis.

Results: We have revised Table 4 based on the updated analysis.

---

## [Decision Letter · Decision Letter 2]

25 Mar 2024

Effects of a dietary intervention on cardiometabolic risk and food consumption in a workplace

PONE-D-23-17601R2

Dear Dr.Archana Shrestha,

We’re pleased to inform you that your manuscript has been judged scientifically suitable for publication and will be formally accepted for publication once it meets all outstanding technical requirements.

Kind regards,

Victor Manuel Mendoza-Nuñez, PhD

Academic Editor

PLOS ONE

---

## [Editor Report · Acceptance letter]

5 Apr 2024

PONE-D-23-17601R2 

PLOS ONE

Dear Dr. Shrestha, 

I'm pleased to inform you that your manuscript has been deemed suitable for publication in PLOS ONE. Congratulations! Your manuscript is now being handed over to our production team.

Kind regards, 

on behalf of

Dr. Victor Manuel Mendoza-Nuñez 

Academic Editor

PLOS ONE